# Virological failure and risk factors among people living with HIV taking second-line ART in Addis Ababa, Ethiopia

**Bekelech Bayou Feyissa**[1,2]*, **Abay Sisay**[3], **Eugene Lee Davids**[1,4], **Anteneh Yalew**[5], **Geoffrey Setswe**[1,6]

1 Department of Health Studies, University of South Africa, Pretoria, South Africa, 2 Deutsche Stiftung Weltbevölkerung (DSW), Ethiopia Country Office, Addis Ababa, Ethiopia, 3 Department of Medical Laboratory Sciences, College of Health Sciences, Addis Ababa University, Addis Ababa, Ethiopia, 4 Department of Psychology, Faculty of Humanities, University of Pretoria, Pretoria, South Africa, 5 Health Sciences Research Office, Faculty of Health Sciences, University of the Witwatersrand, Johannesburg, South Africa, 6 The Aurum Institute, Johannesburg, South Africa

* bekelechbayou@gmail.com

## Abstract

### Background

Virological failure (VF) presents significant challenges in the emergence of drug resistance, and elevated risk of transmission, higher mortality rates, and a diminished quality of life. Various factors contribute to VF, but documented information on this issue is lacking in Ethiopia. Therefore, this study aimed to assess the prevalence of VF and identify the risk factors among people living with HIV who are on second-line antiretroviral treatment (ART).

### Methods

A concurrent mixed-method study using quantitative and qualitative data was conducted at selected hospitals in Addis Ababa, Ethiopia. The analysis was conducted using SPSS version 28, Stata version 18.5, and R for quantitative data and thematic analysis with Atlas.ti version 24 software was used for qualitative data.

### Result

Among 369 adults living with HIV taking second-line ART enrolled in the study, 191 (52%) were male with a median age of 44 years. The prevalence of VF was 55 (14.9%, 95% CI: 11, 19), with an incidence density of 27.2 per 10,000 person months (95% CI 21.1, 35.5). Lost to follow-up significantly increased VF risk [AHR: 2.52 (95% CI: 1.35, 4.69, p-value: 0.004)]. Patients transferred from other health facilities were two times at higher risk of VF compared to those receiving ART at the same facility [AHR: 1.97 (95% CI: 1.07, 3–64, p-value: 0.029)]. Likewise, clients with a history

**Data availability statement:** All relevant data are within the paper and its Supporting information files.

**Funding:** Addis Ababa University supported the research data collection expenses for this study through an adaptive research and problem--solving project, under the grantee Dr. Abay Sisay, with reference number RD/PY-597/2024. Bekelech Bayou Feyissa received support from the University of South Africa (UNISA) through the Bursary System, with reference number 20265433. The funders had no role in the study design, data collection and analysis, decision to publish, or preparation of the manuscript. The content reflects solely the views of the authors and does not represent the opinions of the funders.

**Competing interests:** The authors have declared that no competing interests exist.

of regimen change were at a higher risk of VF [AHR = 2.05, (95% CI: 1.08, 3.88, p-value = 0.027)]. The qualitative data also supported these findings.

## Conclusion

This study underscores the need for improved ART adherence and consistent care to reduce virological failure in PLHIV to improve the quality of life.

## Introduction

Virological failure in people living with HIV (PLHIV) receiving second-line antiretroviral treatment (ART) is defined as two consecutive viral load (VL) test results exceeding 1,000 copies/mL after six months of treatment [1,2]. This occurs when second-line ART fails to suppress HIV replication, preventing the reduction of HIV RNA to undetectable levels, contrary to the primary goal of ART.

PLHIV on ART can present diagnostic challenges. Some may appear clinically stable despite experiencing virological failure, while others may exhibit clinical symptomatic despite having a suppressed viral load. These discrepancies can lead to misdiagnosis, unnecessary treatment switching, and significant implications for individuals, public health, and HIV programs [3].

Virological failure is a major public health concern, contributing to increased HIV transmission rates, drug resistance, opportunistic infections, and treatment challenges [4]. Study done in resource-limited settings revealed second-line VF rates of 21.8%, 23.1%, 26.7%, and 38.0% at 6, 12, 24, and 36 months of ART, respectively [5]. Despite these high rates, second-line ART failure was primarily assessed using clinical and immunological criteria until recently. These assessment criteria are insufficient for detecting early VF, potentially delaying appropriate treatment switches and worsening health outcomes for PLHIV [6].

Even though VL testing is the gold standard for detecting second-line ART failure, in Ethiopia, routine implementation began in 2018 [4]. Studies in Ethiopia reported second-line ART failure rates of 9.86 per 100 person-years and 61.7 per 1,000 person-years in different regions, assessed using both immunological and clinical criteria [7–9]. In 2023, an estimated 39.9 million PLHIV were reported globally, with 38.6 million adults. Approximately 1.3 million new infections occurred, with adults accounting for 1.2 million of these. In 2022, globally, 86% of PLHIV knew their HIV status; of these 77% were on treatment, and 72% of those on treatment achieved viral suppression [10].

Ethiopia had an estimated 610,350 PLHIV, with an overall prevalence of 0.91% [11]. Despite the UNAIDS' 95-95-95 ambitious target to end AIDS as a public health threat by 2030, aiming for 95% of PLHIV to know their status, receive ART, and achieve viral suppression, Ethiopia reported lower performance in this regard [12,13].

Several factors contribute to second-line virological failure, including delayed transition from first-line to second-line ART [14], low CD4 counts (<100 cells/mm³), WHO clinical stage 4 at second-line ART initiation, experienced drug side effects, and

opportunistic infections [15]. PLHIV who experienced VF on first-line ART are also at higher risk of second-line virological failure; 60% of patients with first-line VF developed second-line ART failure [16]. Adherence means consistently following the prescribed medication, and it's essential for effective ART and helps prevent treatment failure [17].

Patients experiencing second-line ART failure switched to third-line regimens, which are complex, expensive, yet highly effective in achieving viral suppression. Studies demonstrate remarkable virological suppression rates of 83% and 93% at 6 and 12 months after switching to third-line ART, respectively [18]. A study in Ethiopia also demonstrated that third-line ART could reduce viral load to undetectable levels [19]. However, access to third-line ART remains limited, with significant delays in patient transitions from second-line to third-line regimens due to various factors. The median time to switch from second-line to third-line ART was 500 days, largely due to delays in requesting drug resistance tests and clinicians' decision-making processes [20].

Despite advancements in HIV treatment, second-line ART failure is a growing concern in Ethiopia, with failure rates reported at 9.86 per 100 person-years in Gondar and 12.2% in Addis Ababa [6,21]. Reliance on clinical and immunological criteria over viral load testing has delayed the detection of treatment failure. Understanding the prevalence of second-line ART virological failure is vital for public health interventions, aiding policymakers in resource allocation and helping healthcare providers identify at-risk patients.

Thus, this study aimed to assess the prevalence and determine the risk factors of second-line ART virological failure among PLHIV in Ethiopia. Specifically, it investigates risk factors influencing VF among adults aged 18 years and older who switched to second-line ART in Addis Ababa from 2018 to 2022. By examining these factors within the Ethiopian healthcare context, it seeks to improve ART monitoring and inform strategies to optimize HIV care and ensure the best quality of life for PLHIV.

## Materials and methods

### Study design, period, and settings

This study utilized a concurrent mixed cohort research design by analyzing both quantitative and qualitative data simultaneously to assess the prevalence of virological failure among individuals aged 18 and older living with HIV (PLHIV) on second-line antiretroviral therapy (ART) enrolled between 2018 and 2022 in Addis Ababa, Ethiopia. The overall research period, which includes both retrospective data summarization, prospective data collection (includes extracting data from the record, FGD, and KII), and analysis, lasted from August 20, 2024, to November 25, 2024. The integration of the findings from both analyses allowed for a comprehensive conclusion through triangulation, S1 Fig.

The study was conducted in selected health facilities in Addis Ababa, the capital of Ethiopia. With a population of approximately 7.8 million, the city's health system includes over 98 government-owned health centers and 13 hospitals, which include Zewditu Memorial Hospital and Yekatit 12 Hospital Medical College, both had a good experience and are pioneers in HIV/AIDS service [22].

As of June 2023, Zewditu Memorial Hospital had provided ART to over 7,632 PLHIV, with 912 on second-line ART, and the Yekatit 12 Hospital Medical College had 3,322 PLHIV on ART, including 453 receiving second-line ART. These hospitals were selected as the study sites.

### Study population and sampling

This study implemented both quantitative and qualitative data components. For the quantitative phase, 369 participants were enrolled using systematic random sampling from the calculated and targeted sample size of 371, proportionally allocated across hospitals based on client load. Data sources were hospital-level medical records of PLHIV, including ART registration, patient charts, and the electronic database.

The qualitative phase employed Key Informant Interviews (KII) involving PLHIV adults taking second-line ART, Focused Group Discussions (FGD) with Hospital-level ART service providers (including clinicians, pharmacy professionals,

adherence supporters, case managers, data clerks), and HIV program managers from the Ministry of Health (MOH), the Addis Ababa City Administration Health Bureau (AACAHB), and sub-city levels.

KII and FGD participants were purposefully selected based on their engagement and expertise in the ART service and program.

The KII participants were selected based on their status as ART service users, and the FGD participants were selected based on their specific professional roles as HIV service providers in the hospitals and program managers.

**Inclusion and exclusion criteria.** Adult PLHIV aged 18 years and older who were receiving second-line ART between 2018 and 2022 for more than six months, had undergone at least two follow-up HIV viral load tests, and had complete medical records were included in this study.

Adult PLHIV aged 18 years and older, who were currently taking second-line ART, had a regular follow-up visit to the hospitals during the data collection period, and willingly provided their informed consent for the interview, have been included in KII. Individuals who were on first-line ART, under 18 years old, or who did not volunteer were excluded from the KII.

HIV service providers (including ART clinicians, pharmacists, case managers, data clerks, and adherence supporters at selected hospitals) and HIV program managers at the regional and sub-city health bureau level were included in the FGD. Individuals who did not hold one of these specific roles were excluded from the FGDs.

### Sampling techniques and sample size determination

**Sample size determination.** To determine the sample size for the quantitative study, the following assumptions were considered, taking the 12.22% prevalence of second-line ART virological failure from a study conducted in Addis Ababa [6], a precision (D) of 3.5%, and a CI of 95%, the sample size for the study after adding 10% contingency using below formula became 371.

$$n = Z^2 \alpha/2 \, p(1-p)/d^2$$

Z= 1.96 with 95% CI, P= 12.22%, d= 0.035.

For the qualitative component of this study, non-probability purposive sampling was used to recruit participants based on their active engagement and roles in the HIV program, with their voluntary participation. Pertinent information was collected from participants using the Key Informant Interview (KII) and focus group discussion (FGD) guide until data saturation was achieved. Data saturation refers to the point in data collection and analysis when new data no longer yields significant or novel insights or themes, indicating that theoretical saturation has been achieved [23]. As there is no similar research conducted in Ethiopia, the researcher used a similar study conducted in Uganda as a reference [24]. In this study, twenty (20) PLHIV taking second-line ART were interviewed using a key informant interview guide. Participants who met the inclusion criteria were selected purely for the KII and assigned to the respective hospitals where the research was conducted.

In addition to the KII, the researcher has conducted three FGDs to explore the factors that affect the adherence of PLHIV aged 18 years and older taking second-line ART in Addis Ababa, Ethiopia. The three FGDs encompass: two with hospital-level ART service providers and one with the HIV programme managers of AACAHB and sub-cities. It was composed of five to seven individuals. Focus group participants were invited to be part of the study based on the inclusion criteria after giving their written consent.

To safeguard the credibility of results, observation and field notes were utilized. The interviews and FGDs were recorded as this helped the researcher to develop notes, which assisted during the analysis of codes.

**Sampling techniques.** A systematic random sampling (SRS) technique was applied using probability proportional allocation. As illustrated in Fig 1. Accordingly, 249 samples were allocated to Zewditu Memorial Hospital, while 122 were assigned to Yekatit 12 Hospital Medical College.

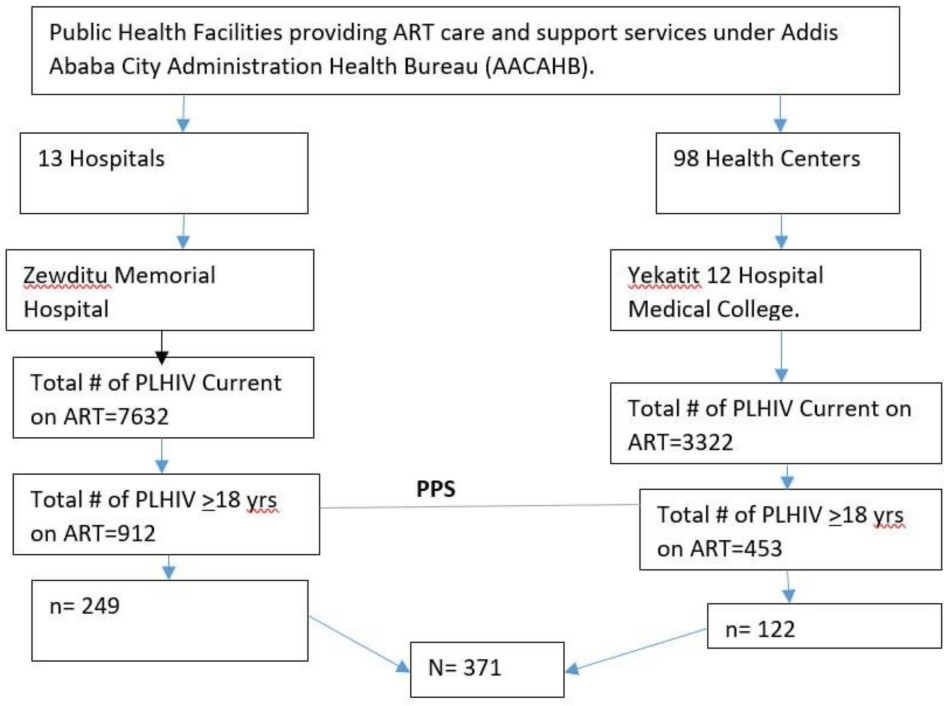

*PPS-population proportion sampling*

**Fig 1. The population proportion sampling technique used in this study.**

## Data collection method and quality assurance

Data were collected using a structured checklist that contained a comprehensive list of required variables addressing the study's objectives. The checklist was pretested with 5% of the total data, and this pretested data was not included in the final analysis. This pretesting allowed for necessary revisions and improvements to the final data collection tool, ensuring that no essential data was overlooked. To facilitate the data collection process and enhance data quality monitoring, the researcher converted the questionnaires into the Kobo Toolbox app, a free, cloud-based data collection tool that allows for real-time research data gathering.

In each hospital, one supervisor and two data collectors were recruited based on their work experience and educational background. They are senior public health professionals who have recently received training on the consolidated guidelines for HIV ART and are actively involved in the program. To ensure the reliability of the quantitative data, both data collectors and supervisors received one day of training to familiarize them with the data collection tools. Additionally, supervisors and the researcher spent extra time closely monitoring the data collection process to ensure the credibility of the data. The data collection tools were pretested to identify and incorporate any missing variables, and regular data quality checks were conducted. All supervisors and data collectors involved in this research signed a confidentiality agreement.

For the qualitative part, information was collected using the KII guide. The guides were initially prepared in English and then translated into Amharic (the local language) to gather complete information, which was subsequently transcribed back into English [25]. Written informed consent was obtained from the key informants, who participated voluntarily.

In each hospital, one supervisor and one interviewer were selected for qualitative data collection, based on their work experience and educational background. These individuals received a one-day orientation on the interview guide

to ensure consistent understanding and maintain data quality. The checklists and interview guides were pretested, and appropriate revisions were made before the actual data collection.

To ensure consistency, KII and FGD guiding questions were developed, and all data collectors were oriented on these questions before data collection commenced, ensuring the information was gathered accordingly.

The key topics covered during the interviews and discussions centered on developing effective and targeted strategies to improve ART adherence and achieve optimal health outcomes for PLHIV taking second-line ART.

During the study, the plan was to conduct interviews with twenty (20) people living with HIV (PLHIV) who were receiving second-line ART. However, data saturation was achieved with 15 interviews (75%) completed using the KII guide. Participants who met the inclusion criteria were selected voluntarily and assigned to the hospitals where the research was conducted. To mitigate potential selection bias, participants were recruited based on a pre-tested standardized inclusion criterion by a trained research data collector, with oversight from the study supervisor to ensure procedural fidelity. Additionally, the researcher conducted FGDs using the FGD guide. Three FGDs were held: two with hospital-level ART service providers and one with HIV program managers from AACAHB and sub-cities, composed of five to seven individuals each. Focus group discussants were invited based on the inclusion criteria, and they signed a written consent form for participation.

## Data analysis

Quantitative data were analysed using SPSS version 28, Stata version 18.5, and R. Descriptive statistics estimated the proportion of virological failure in patients undergoing second-line ART. Cox regression was utilized to control for confounding factors and identify independent predictors of treatment outcomes. A survival analysis using the Kaplan-Meier estimator examined the time to virological failure, while a life table analysis assessed the probability of failure at various intervals. Factors with a p-value less than 0.20 in bivariate regression were included in a multivariate model, with significant results defined as $p < 0.05$.

The qualitative data analysis was conducted using a thematic analysis approach, facilitated by Atlas.ti software, Version 24. The qualitative data from the KIIs and FGDs were imported into Atlas.ti software, where a systematic thematic analysis was conducted by grouping the documents, generating initial codes, refining and grouping the coded segments, and then performing final analysis and visualization. Hence, the themes were generated directly from the data without trying to fit them into pre-existing theory or framework. The specific theme the researcher used was the inductive thematic analysis.

The findings from both the quantitative and qualitative analyses were integrated and triangulated to develop a more comprehensive conclusion and formulate effective recommendations.

The quantitative and qualitative data were analyzed separately using SPSS software version 28 and ATLAS.ti software version 24, respectively. Key findings from both datasets were converted into a common format. Quantitative results were presented as statistical outcomes, while qualitative themes were converted into descriptive statements. The results were systematically compared on a point-by-point basis to identify areas of convergence, divergence, and expansion. The final discussion and conclusion synthesized these compared findings.

The overall summary of defining virological failure in the research process is summarized and depicted using the infographic in S2 Fig.

## Ethical consideration

This research underwent comprehensive ethical approval processes, received appropriate ethical clearance from UNISA (CREC Reference #: 20265433_CREC_CHS_2024), the Addis Ababa Health Bureau (A/A/14641/227), and two hospitals: Zewditu Memorial Referral Hospital and Yekatit 12 Hospital Medical College (reference #: 439/24). Approval was secured prior to data collection, with written consent obtained from ART clinic focal persons and hospital managers. Participant confidentiality and anonymity were prioritized, with voluntary participation and the freedom to withdraw without affecting care.

The study adhered to ethical principles of autonomy, beneficence, justice, and non-maleficence. Confidentiality was ensured through secure storage, coded identifiers, and signed agreements from data collectors. For the qualitative part, purposive sampling was used to ensure equal representation of genders, while the quantitative study employed systematic random sampling. No interventions or additional risks were posed to participants.

## Results

### Baseline second- line ART regimen dispensed for the study participants

In this study, out of the calculated sample size of 371, a total of 369 individuals living with HIV (PLHIV) who were receiving second-line antiretroviral therapy (ART) were enrolled, representing a 99.5% response rate of the calculated sample size. Among the participants, taking the prescribed second-line ART drugs, 2h and 2f were the most frequently dispensed drugs among adult HIV-positive patients experiencing second-line ART regimen failure at selected hospitals in Addis Ababa, 2018–2022 (n = 369), Table 1.

### Reasons for second-line ART regimen change

Among second-line ART regimens, changes may occur for various reasons, including treatment failure, adverse drug effects, and poor adherence. Additionally, during treatment follow-up, clients may experience regimen changes even without encountering treatment failure. To optimize treatment effectiveness, the choice of second-line therapy often considers the latest clinical advancements and the client's first-line regimen history. While not highly recommended, clients might change their medications after starting a regimen, especially in resource-limited settings, to improve quality of life.

In the follow-up of HIV treatment, there could be various reasons for changes in the treatment regimen among clients receiving second-line ART. In this study, of the 369 enrolled HIV-positive clients on second-line ART, 72 (20%) had well-documented ART regimen changes, Fig 2A. The most common reason for these changes was drug toxicity or side effects, 47 (65%), followed by the availability of new drugs at healthcare facilities, 11 (15%), as illustrated in Fig 2B.

The pie chart (Fig 2A) illustrates the extent of regimen changes among sampled second-line ART treatment failures in adult HIV-positive patients at selected hospitals in Addis Ababa, Ethiopia, from 2018 to 2022 (n = 369).

To mitigate adverse ART outcomes and address adherence challenges related to drug side effects, appropriate ART regimen changes are essential. Such modifications are crucial for improving overall treatment outcomes and preventing long-term health complications. In resource-limited settings, where alternative drugs may not always be readily available, careful clinical assessment and prioritization of tolerable yet effective treatment options are critical to maintaining viral

**Table 1. Type of second-line ART regimen taken among PLHIV in selected Hospitals in Addis Ababa, Ethiopia, 2018-2022 (N = 369).**

| Type of second- line regimen | Frequency | Percent |
|---|---|---|
| Other adult second-line | 82 | 22.2 |
| 2e (AZT + 3TC + LPV/r) | 16 | 4.3 |
| 2f (AZT + 3TC + ATV/r) | 83 | 22.5 |
| 2g (TDF + 3TC + LPV/r) | 33 | 8.9 |
| 2h (TDF + 3TC + ATV/r) | 132 | 35.8 |
| 2i (ABC + 3TC + LPV/r) | 22 | 6.0 |
| 2k (AZT + 3TC + DTG) | 1 | 0.3 |
| | **369** | **100** |

**A**

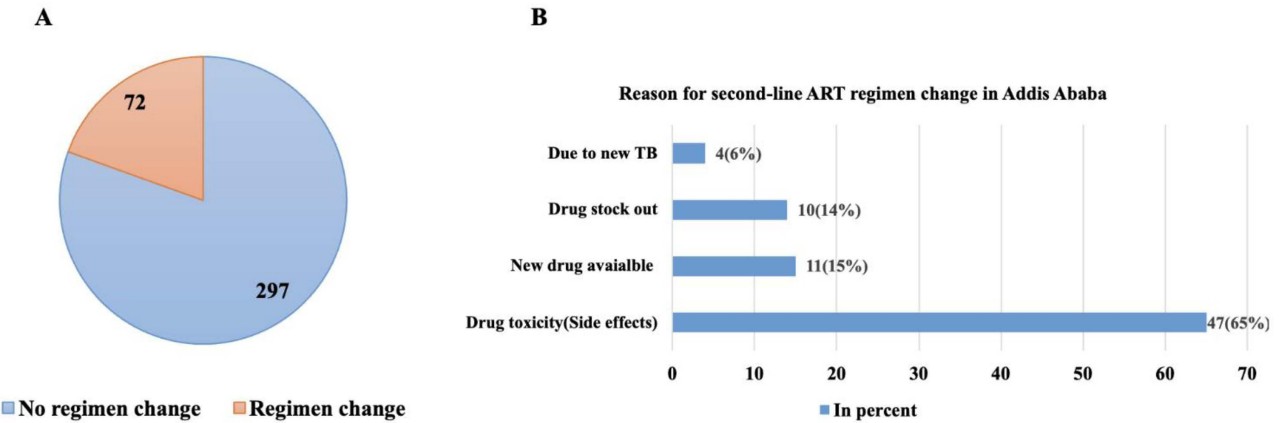

72

297

■ No regimen change  ■ Regimen change

**B**

Reason for second-line ART regimen change in Addis Ababa

Due to new TB  4(6%)
Drug stock out  10(14%)
New drug avaialble  11(15%)
Drug toxicity(Side effects)  47(65%)

0  10  20  30  40  50  60  70

■ In percent

**Fig 2. Second-line ART regimen change among PLHIV in Addis Ababa, Ethiopia.** (A) Second-line ART regimen change experience among enrolled patients in Addis Ababa, Ethiopia. (B) Reason for a second-line ART regimen change among PLHIV in Addis Ababa, Ethiopia.

suppression and preventing ART failure. In this study, ART side effect or drug toxicity is the top reason for second-line regimen change, as indicated in Fig 2B.

## Opportunistic infections among second-line ART

The role of opportunistic infections (OIs) in the progression of HIV and the decline in the quality and effectiveness of treatment is a critical issue that should not be overlooked. Their impact on the success of treatment programs is significant. In this study, we examined the prevalence of OIs among HIV-positive individuals undergoing second-line ART. Out of 369 participants, 48 clients developed OIs during their second-line treatment. Tuberculosis was the most common, 19 (39%), followed by herpes zoster, as shown in Fig 3.

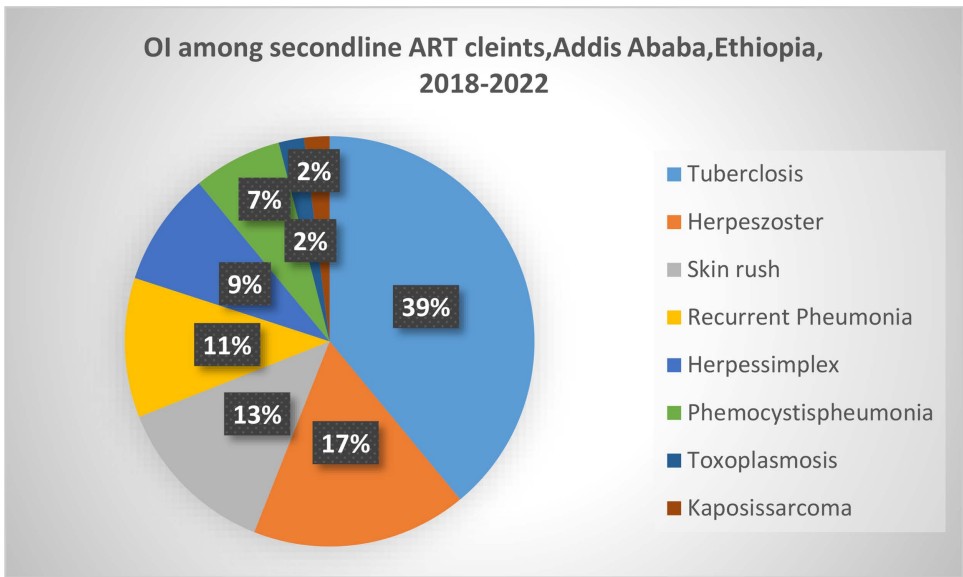

**Fig 3. Type of OI developed among PLHIV taking second-line ART in selected Hospitals in Addis Ababa, Ethiopia, 2018-2022.**

## Time to OI develop in PLHIV taking second-line ART

Recent evidence-based approaches to disease management emphasize the importance of ensuring a high quality of life for HIV patients receiving second-line ART, which requires effective management of opportunistic infections throughout their treatment. In this study, the researcher got a chance to evaluate the prevalence and timing of opportunistic infections development among the 369 participants. Notably, clients in the 24–30 month treatment window exhibited the highest incidence of opportunistic infections, as illustrated in S3 Fig.

## Magnitude and risk factors associated with VF in second-line ART

In this study, a total of 369 individuals were enrolled, and 55 of them (14.9%, 95% CI 11, 19) experienced second-line ART VF with 20,187 months of time at risk. The risk of virological failure in clients receiving second-line ART was found to be higher among those who came from other health facilities, with an adjusted hazard ratio (AHR) of 1.97 (95% confidence interval (CI): [1.07, 3.64], P-value = 0.029). Second-line ART regimen change history was also found to be a predictor of second-line ART VF (AHR = 2.05, 95% CI: [1.08, 3.88], P-value = 0.027). The other predictor found in this study was a history of LTFU from ART treatment is 2.5 times more likely to have VF than those retained in care (AHR = 2.52, 95% CI: [1.35, 4.69], P-value = 0.004), as the details depicted in Table 2.

## Time to treatment failure among second-line ART patients

The patients were followed for a minimum of 20 and a maximum of 74 months. Out of 369 study participants followed, a total of 55 (14.9%, 95% CI 11, 19) ART clients developed second-line treatment failure in 20,187 person-months of the total analysis time at risk and under person per month (PM) of observations. The incidence density was 2.72 per 1000 PM with a 95% CI of [2.11, 3.55] or 33 per 1000 person-years (PY) with a 95% CI of [25.0, 42.0].

The cumulative survival probability of the sampled client within the cumulative probability of failure at 24 months was 4.9% (95% CI 3.1% to 7.6%). The cumulative survival probability for second-line ART treatment during the follow-up period was found to be decreased, which means, as the time of follow-up goes, the failure rate increases. As depicted in the Kaplan-Meier failure curve, Fig 4A. Kaplan-Meier failure curve showing the last time of failure or event was at 48 months of follow-up from treatment failure of HIV-positive adults on second-line antiretroviral therapy at selected hospitals in Addis Ababa, Ethiopia, 2018–2022,(n = 369) with 95% CI, Fig 4B.

In another way, considering the combined result, the cumulative probability of failure at 24 months was 4.9% (95% CI 3.1% to 7.6%), at 48 months it was 34.2% (95% CI 29.6% to 39.3%), at 60 months it was 80.7% (95% CI 76.6% to 84.6%), and at 72 months it was 92% (95% CI 88.8% to 94.4%), (**Table 3**).

In this study, the occurrence probability of virological failure was detected at 24, 34, 36, 43, and 48 months of treatment follow-up among adults receiving second-line antiretroviral therapy, with the highest incidence noted at 36 months during the follow-up period. These findings highlight the importance of closely monitoring patients during the earlier stages of follow-up, particularly within the first 48 months. Early and regular follow-up efforts are critical, as they help to identify and address potential challenges promptly. After this period, patients tend to achieve greater stability in their treatment regimen, which ultimately could lead to better treatment outcomes and improved overall quality of health (**Fig 4B** and Table 3).

## Results from the qualitative data sources

In this section, the investigators present results obtained from qualitative data sources: from KII and FGD, presented in chronological order

**Bio-data of participants.** In the KII with 15 participants (10 females and 5 males), adults living with HIV who are taking second-line antiretroviral therapy (ART) at Zewditu Memorial Hospital and Yekatit 12 Hospital Medical College participated. The average age of the KII participants was 35.4 years, ranging from a minimum of 24 years to a maximum of 47 years.

**Table 2. Multivariate Cox regression analysis of predictors of second-line ART failure of adult HIV-positive patients at selected hospitals in Addis Ababa, Ethiopia, 2018-2022, (n = 369).**

| Variable | Rx failure | | Crude HR (95% CI) | P-value | Adjusted HR (95% CI) | P-value |
|---|---|---|---|---|---|---|
| | Yes N (%) | No N (%) | | | | |
| **Marital Status** | | | | | | |
| Married | 22 (18) | 101 (82) | 1 | | | |
| Never married | 25 (14) | 160 (86) | 0.76 (0.43–1.36) | 0.364 | | |
| Others [a] | 8 (13) | 53 (87) | 0.71 (0.32–1.61 | 0.424 | | |
| **Sex** | | | | | | |
| Male | 24 (13) | 167 (87) | 1 | | | |
| Female | 31 (17) | 147 (83) | 1.46 (0.85–2.49) | 0.164 | | |
| **Second-line ART regimen change experience** | | | | | | |
| Yes | 17 (4.6) | 55 (14.9) | 1.92 (1.08–3.41) | 0.025* | 2.05 (1.08–3.88) | 0.027** |
| No | 38 (10.3) | 259 (70.2) | 1 | | | |
| **Educational Status** | | | | | | |
| No formal education | 3 (0.8) | 28 (7.6%) | 1 | | | |
| Primary | 23 (6.2) | 84 (22.8) | 2.43 (0.73–8.09) | 0.148 | | |
| Secondary | 22 (6) | 168 (45.5) | 1.25 (0.37–4.17) | 0.718 | | |
| Tertiary | 7 (1.9) | 34 (9.2) | 1.89 (0.48–7.30) | 0.357 | | |
| **Patient linked from** | | | | | | |
| Intra facility | 255 (69.1) | 38 (10.3) | 1 | | | |
| Other facility | 59 (16) | 17 (4.6) | 1.82 (1.027–3.22) | 0.040* | 1.97 (1.07–3.64) | 0.029** |
| **Weight at start of second-line ART** | | | | | | |
| 30-40 | 4 | 15 | 1 | | | |
| 41-50 | 20 | 71 | 0.93 (.31–2.72) | 0.897 | 1.24 (0.41–3.73) | 0.700 |
| 50+ | 31 | 228 | 0.48 (0.17–1.38) | 0.178 | 0.75 (0.25–2.19) | 0.59 |
| **Disclosure** | | | | | | |
| Yes | 52 (14.1) | 257 (69.6) | 1 | | | |
| No | 57 (15.4) | 3 (0.8) | 3.60 (1.12–11.54) | 0.031* | | |
| **Previous TB** | | | | | | |
| Yes | 5 | 30 | 0.98 (0.39–2.46) | 0.972 | | |
| No | 50 | 284 | 1 | | | |
| **T staging at the switch/ at the start of second-line ART** | | | | | | |
| I | 313 | 46 | 1 | | | |
| II | 6 | 1 | 8.65 (3.67–20.37) | <0.001* | 4.84 (1.62–14.41) | 0.005** |
| III | 2 | 0 | 8.52 (2.06–35.16) | 0.003 | 24.34 (1.78–331) | 0.017** |
| IV | 1 | 0 | 10.65 (1.45–77.95) | 0.020 | 487 (3.64–65036) | 0.013** |
| **Adherence Level at the switch/the start of second- line ART** | | | | | | |
| Good | 20 | 142 | 1 | | | |
| Poor | 35 | 172 | 7.51 (4.19–13.44) | <0.001* | 0.98 (0.52–1.83) | 0.949 |
| **Functional Status at the switch/ at the start of second-line ART** | | | | | | |
| Working | 47 | 265 | | | | |
| Ambulatory | 3 | 12 | 1.28 (0.59–2.76) | 0.535 | | |
| Bedridden | 5 | 36 | 1.41 (0.58–2.24) | 0.699 | | |
| **Type of second-line ART dispensed** | | | | | | |
| 2e (AZT + 3TC + LPV/r) | 2 (12) | 14 (88) | 0.84 (0.187–3.74) | 0.817 | | |
| 2f (AZT + 3TC + ATV/r) | 10 (12) | 73 (88) | 0.84 (0.36–1.95) | 0.688 | | |

*(Continued)*

Table 2. (Continued)

| Variable | Rx failure | | Crude HR (95% CI) | P-value | Adjusted HR (95% CI) | P-value |
|---|---|---|---|---|---|---|
| | Yes N (%) | No N (%) | | | | |
| 2g (TDF + 3TC + LPV/r) | 5 (15) | 28 (85) | 1.01(.35–2.86) | 0.985 | | |
| 2h (TDF + 3TC + ATV/r) | 19 (15) | 113 (85) | 0.99(0.48–2.04) | 0.984 | | |
| other adult second- line | 12 (15) | 70 (85) | 2.32 (0.92–5.91) | 0.076 | | |
| Others [b] | 7 (30) | 16 (70) | | 1 | | |
| **ART drug Side effect** | | | | | | |
| Yes | 9 (20) | 34 (80) | 1.55 (0.76–3.17) | 0.227 | | |
| No | 46 (14) | 280 (86) | | | | |
| **Lost Follow Up (LTFU)** | | | | | | |
| Yes | 28 (7.6) | 55 (14.9) | 3.81 (2.24–6.46) | <0.001* | 2.52 (1.35–4.69) | 0.004** |
| No | 27 (7.3) | 259 (70.2) | 1 | | | |
| **OI Prophylaxis (taking CPT….** | | | | | | |
| Yes | 50 (17) | 251 (83) | 1 | | | |
| No | 5 (7) | 63 (93) | 0.43 (0.17–1.10) | 0.080 | | |

* Significance at CHR, ** Significance at AHR

**A**

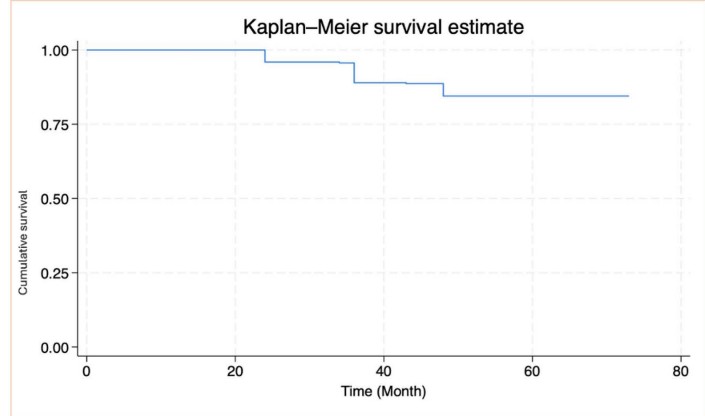

**B**

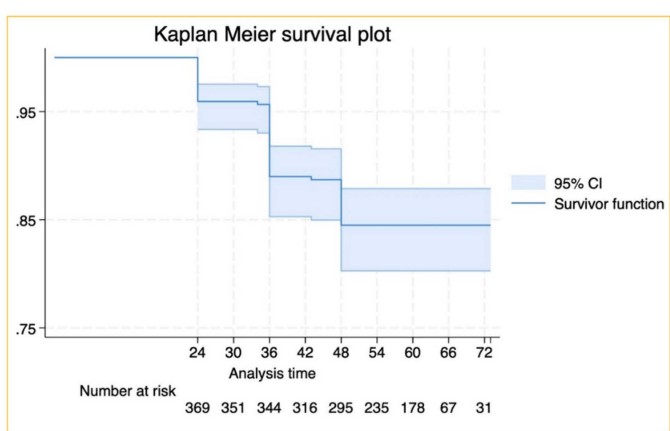

**Fig 4. Kaplan-Meier survival estimates.** (A) Overall Kaplan-Meier failure curve showing hazard from treatment failure of HIV-positive adults on second-line antiretroviral therapy at selected hospitals in Addis Ababa, Ethiopia, 2018-2022, (n = 369). (B) Kaplan-Meier failure curve with 95% CI for treatment failures in HIV-positive adults on second-line antiretroviral therapy in Addis Ababa, Ethiopia (2018-2022, n = 369).

For the FGDs, three discussion sessions were conducted: two at the hospital level with ART service providers and one with program managers. Each FGD consisted of 6–7 individuals. In total, 13 individuals participated in the FGDs, which encompassed three ART focal persons, 2 were female and ART providers (two female and one male included), three adherence counselors (one female), and four case managers (one male).

In the FGD with program managers, a total of 6 individuals participated (5 females and 1 male).

**Reasons for virological failure from key informant interviews.** In this study, the plan was to conduct interviews with twenty (20) people living with HIV (PLHIV) taking second-line antiretroviral therapy (ART), aged 18 years and above.

**Table 3. Cumulative probability of failure of adult Patients on second-line ART at selected hospitals in Addis Ababa, Ethiopia, 2018-2022.**

| Follow-up time in Months | Cumulative Probability of failure(in percent with 95% CI) |
|---|---|
| 24 | 4.9 (3.1,7.6) |
| 30 | 5.2 (3.3,7.9) |
| 36 | 14.4 (11.2, 18.4) |
| 42 | 14.6 (11.4, 18.7) |
| 48 | 34.2 (29.6,39.3) |
| 54 | 36.9 (32.2, 42.0) |
| 60 | 80.7 (76.6, 84.6) |
| 66 | 82.7 (78.6, 86.3) |
| 72 | 91.9 (88.8, 94.4) |

However, data saturation was achieved with 15 (75%) interviews completed, using a key informant interview guide. Participants who met the inclusion criteria were selected voluntarily from the two selected hospitals.

These sessions were facilitated by a trained facilitator, along with a note-taker, using a pre-set script or guide questions. The discussions were audio-recorded, then transcribed and translated into written transcripts. The transcripts were analyzed using the qualitative analysis software ATLAS.ti V. 24, identifying five themes, networking was done, and the findings were presented in a summarized manner in these five themes of categories or sub-themes. The researcher presented the findings accordingly.

| Themes | Sub-themes |
|---|---|
| Drug-related | • Frequency of taking medication<br>• Challenges faced in taking medication<br>• Feelings about the long-term commitment to taking second-line antiretroviral therapy |
| Disclosure-related | • Feelings about disclosing your status to others<br>• Challenges in disclosing HIV status<br>• Benefits and risks of disclosing HIV status |
| Social support-related | • Support groups help in taking ART<br>• What participants like to see in a support group or organization for PLHIV |
| Distance from HF | • Ever missing a dose of ART drugs because of difficulty getting to the HF |
| Stigma and discrimination-related | • Effect of stigma or discrimination on adherence to ART<br>• Coping with stigma and discrimination |

Following the exchange of ideas and discussions of each theme and sub-themes, the researcher explained, and the results are presented.

**Drug-related: Pill burden, timing, and drug side effects:** *"…taking medication for such a long period is very tough and sometimes feels boring." "… There are days when I feel too tired or overwhelmed by the daily routine of taking my medication, but I remind myself of its importance". Sometimes, I forget or get caught. Second-line ART is more challenging than first-line treatment due to the increased number of pills, leading to treatment fatigue.*

*…, however, the pill burden is challenging me. I sometimes worry about the potential long-term side effects of the drug… taking medication for the long term is so difficult…"*

*"There have been times when I have missed doses of my ART during social activities because I sometimes feel uncomfortable taking them in front of others during social activities due to fear of stigma. These situations have led to missed doses, which I know can affect my health".*

*"I used to miss doses, and I experienced the negative impact it had on my health."*

*"…..side effects like insomnia and trouble sleeping, which have been affecting my mood and energy level, making it harder to perform my duties at full capacity"*

Inconsistent medication adherence is often due to fear of stigma, leading individuals to conceal their HIV status and experience isolation. This lack of support increases stress and disrupts routines. *".. There are times when I miss or delay doses because of the strict schedule. My fear of stigma and discrimination prevents me from openly discussing my HIV status, even with those close to me. This secrecy means I don't have a support system to remind or encourage me to take my medication consistently, which contributes to my inconsistent adherence. I feel isolated in managing my condition, and the constant worry about others discovering my status adds to my stress, making it even harder to maintain a regular medication routine."* [Male KII participant]

**Disclosure-related:** One of the interviewees also stated that stigma is a barrier to getting support from others. Participants in FGDs raised concerns about stigma and discrimination. One of them said:

*"Many PLHIV are afraid of stigma and discrimination, so they choose to keep their status a secret. Because of this, they miss out on the help they could get from their family, partner, and colleagues, which could have made it easier for them to follow their treatment properly."* [FGD participant]

This is supported by ART providers and witnessed with cross-referencing to the FGD participants' thoughts from the Hospitals identified economic and financial difficulties as a challenge in treatment adherence.

Fear of social stigma and discrimination is a major barrier to treatment adherence. *"…for many patients, economic difficulties and financial instability make it hard to stay adherent. Some clients travel long distances, even from regional states, to access their ART due to fear of social stigma and discrimination. This creates a significant barrier, as the long journey adds transportation challenges, making it harder for them to consistently adhere to their treatment."* [ART provider]. *Disclosing my HIV status to others has never been an easy decision. I know that opening up about it could expose me to judgment and stigma, which is something I deeply fear. I worry that people might look at me differently, treat me unfairly, or even reject me entirely. So far, the only person who knows about my HIV status is my father. I trust him completely, and I feel safe sharing this part of my life with him. Beyond that, I have no desire or plan to tell anyone else."* [FGD participant]

A witness from another key informant,*"…the only person that knows my HIV status is my mother. My HIV status is known only to my mother, and I have chosen to keep it confidential from other people. Disclosing HIV status can sometimes lead to stigma and discrimination, and I want to prioritize protecting myself from those potential experiences by not disclosing my HIV status."* [Female KII participant]

From other participants, *"…Since the community's awareness about HIV is low, I avoid carrying my medication to school or the workplace to prevent being seen and possibly stigmatized or discriminated against. Instead, I ensure that I stay at home during my medication time to avoid missing a dose. This decision stems from a fear of judgment and the negative reactions I might face if someone discovers my HIV status. While this approach helps me adhere to my treatment schedule, it limits my flexibility and affects my daily routine, as I have to plan my activities around being at home for my doses. I hope that with increased community awareness and education about HIV, I can take my medication confidently, no matter where I am, without fear of stigma or discrimination."* [Male KII participant]

*"…I am not willing to disclose my status to anyone in the future due to fear of stigma and discrimination."* [KII participant]

**Social support-related:** Despite advancements in the HIV/AIDS program, many social support groups and non-governmental organizations' activities contribute to supporting the program and facilitating service utilization. However, the

majority of the Key Informant Interview (KII) responses do not positively align with fostering a seamless flow of information. Instead, they express suspicions towards service providers and highlight perceived stigma.

*"…I am not currently participating in any social support groups. My decision stems from concerns about privacy and confidentiality, as I fear that joining such groups might inadvertently disclose my HIV status to others. Additionally, I worry about facing stigma and discrimination, which are prevalent in my community. However, I acknowledge that support groups can offer benefits, such as emotional support, shared experiences, and practical advice on adhering to antiretroviral therapy. If I were to consider joining a support group, I would find it most beneficial if the group ensured strict confidentiality, provided a non-judgmental environment, and offered flexible meeting times to accommodate my schedule."* [Male KII participant]

*"….I haven't joined any social support groups. It's not that I don't see their value; However, I have some reservations. I am concerned about my privacy and the possibility of unintended disclosure of my HIV status, which could lead to stigma or discrimination"* [Male KII participant]

**Distance from Health Facility:** "I am living out of Addis Ababa, and sometimes I am encountering difficulty in maintaining my appointment date."
Thoughts from other participants, who have not missed their treatment:

*"I have never missed a dose of ART due to difficulties in reaching the health facility. However, I have missed doses because of the stigma associated with HIV. The fear of others discovering my status makes me hesitant to take my medication in public or even discuss my condition, leading to missed or delayed doses. This fear of stigma and discrimination has a significant impact on my adherence to the treatment regimen."* [Male KII participant]

**Stigma and discrimination-related:** Many PLHIV are afraid of stigma and discrimination, so they choose to keep their status a secret.

*"… address stigma and discrimination and improve patient adherence to ART, community awareness efforts must be strengthened. HIV awareness has declined in recent years, with the declining number of media coverage and growing misconceptions in the community, such as the belief that HIV has been eliminated."* [KII participant]

The other participant said, *"While there has been some improvement in reducing social stigma and discrimination against PLHIV in the community, challenges still persist. Many PLHIV continue to face stigma, with some members of the community perceiving them as 'ill-mannered'. To truly address this issue and improve patient adherence to ART, a multi-sectorial collaboration approach is essential to create a more supportive and inclusive environment for PLHIV, reduce misconceptions, and foster a culture of respect and acceptance."* [KII participant]

*"…I've seen how society can stigmatize people living with HIV, and I want to protect myself from that experience." "…I didn't want anyone to know, and this made it especially hard to take my medication in public or around people I knew."* [Male KII participant]

**Factors influencing VF to second-line ART from focus group discussions.** Understanding the factors for VF to second-line ART is crucial for improving treatment outcomes and ensuring the long-term success of HIV programs. To gain insights from a programmatic perspective as a way of "hearing from the horse's mouth," three different focus group discussion (FGD) sessions were conducted with 5–7 participants each, including ART service providers and HIV program managers working at the regional health bureau, sub-city health department, and two hospital levels. The findings

presented in the following sections offer a comprehensive overview of the ART program from both a service delivery and programmatic standpoint, highlighting practical, scalable solutions to strengthen support and improve treatment outcomes for PLHIV on second-line ART.

The FGD participants identified key barriers mainly associated with psychological, emotional, psychosocial, cultural, and religious factors, explored successful strategies, and proposed innovative solutions to enhance patients' ART outcomes. For simplicity and clarity, the researcher presented the findings accordingly.

**Successful strategies to improve adherence to ART:** The study also explored successful strategies that have been implemented to improve patient outcomes with ART. Participants highlighted several effective approaches, including: "…*the Differentiated Service Delivery (DSD) model, Youth Program, and Three-Month Multi-Month Dispensing (3MMD), which help reduce the burden of frequent hospital visits. Additionally, Enhanced Adherence Counseling (EAC), pill counts, and adherence supporters have played a crucial role in monitoring and encouraging patients to stay on treatment*" [FGD participant].

FGD participants proposed specialized care and follow-up services. One of them said:

"*For patients with Advanced HIV Disease (AHD), specialized care and follow-up services have been established to ensure better health outcomes. Moreover, Community ART Groups (CAGs) and Peer Community ART Groups (PCAGs) have proven to be highly effective, as they provide peer support and motivation. These strategies, implemented by the government and development partners, have significantly contributed to improving ART adherence*" [FGD participant].

One of the FGD participants said:

"---*the 3MMD model allowing stable patients to receive three months' worth of medication at once, reducing frequent hospital visits*." An adherence supporter said, "---*We've also seen positive outcomes from Enhanced Adherence Counseling, where we provide continuous support for patients struggling with adherence, and Pill counts help us monitor whether patients are taking their medication correctly*." [FGD participant]

ART provider also added,

"--- the *CAG program facilitates patients' support for each other in managing their treatment, and similarly, the PCAG has been very effective as patients find it easier to stay on ART when they are encouraged by others who share the same experience*." [ART provider]

**Challenges associated with implementing the best identified Strategies:** The study also looks at the challenges associated with implementing the above-mentioned strategies at the hospital and programmatic levels. Accordingly, the FGD participants noted that despite the effectiveness of these strategies, some programs are regressing due to inadequate funding for the ART program and shortages of second- and third-line ART.

One of the ART focal said, "---*Due to the shortage of second-line ART drugs, we are unable to implement the 3MMD DSD model. Currently, we only provide second-line ART on a monthly basis. If we had an adequate supply of these medications, we preferred and could extend patients' appointments to two or three months, making it possible to implement the 3MMD DSD model fully*."

The study also explored the required resources to improve PLHIV adherence. Accordingly, FGD participants raised adequate human resources, improved infrastructure, uninterrupted ART drug supply, financial support, and nutrition support.

One of the hospital ART focal persons said,

"---*We need adequate and skilled healthcare providers to deliver comprehensive and quality adherence counseling and follow-up services.*" The other adherence counselor said, "---*private adherence counseling rooms foster trust and uphold privacy, allowing patients to openly discuss their concerns without fear of stigma or judgment.*" She also added, "--- *Improved infrastructure like adequate and comfortable waiting areas equipped with TV screens can improve patient engagement and education by providing valuable information about HIV and ART adherence.*" [Hospital ART focal person]

In relation to second-line ART supply, one of the ART providers said,

"--- *giving patients a longer refill period, like the 3-month multi-month dispensing (3MMD), really helps. It reduces the number of visits to the hospital, which is important for those who live far away or have work commitments. When patients do not have to come to the Hospital every month just for a refill, they are more likely to continue taking their medication properly. It also reduces the workload for health workers, allowing them to focus on those who need more attention, like new patients or those struggling with adherence.*" [ART provider]

Concerning nutrition, a case manager says, "---*there are patients who experience food insecurity, with some even requesting food support from us. When patients lack proper nutrition, the risk of treatment interruption increases. Providing nutrition support, whether through direct food assistance or linkage to social programs, can play a crucial role in improving adherence rates and enhancing overall patient well-being.*" [Case manager]

**Use technology to improve patient adherence to ART:** Furthermore, the study examined technology-driven initiatives aimed at improving ART adherence among People Living with HIV (PLHIV). FGD participants highlighted the use of Short Message Service (SMS), particularly for patients with a high viral load, as well as phone alarm reminders to support medication adherence. Additionally, the implementation of the SMART Care electronic medical record initiative was emphasized as a critical strategy for tracking missed appointments and identifying patients lost to follow-up.

To further strengthen adherence support, a hospital adherence supporter recommended introducing an online system for real-time high viral load test notifications and remote adherence counseling. She added that such a system would ensure timely interventions and enhance ART adherence.

**Overall effectiveness of the ART program:** In the current extreme variability and scarcity world, having an effective and efficient ART program is crucial for sustainably controlling HIV/AIDS. It significantly lowers viral loads, reduces the occurrence of opportunistic infections, and enhances the quality of life and survival rates for individuals living with HIV. Accordingly, the study assessed the overall effectiveness of the ART program from the perspective of FGD discussants. The FGD participants noted that this has been evaluated through management reviews, periodic program performance assessments, catchment area meetings, and joint supportive supervision.

As a result, a program expert stated, "The success of the ART program has played a crucial role in saving the lives of PLHIV. Mortality due to HIV has significantly decreased compared to previous years, which indicates the effectiveness of the program." Another ART provider also emphasized, "We can measure the success of the ART program through viral load results. The viral load suppression rate is close to 95%, demonstrating that the ART program has been highly successful."

## Discussion

Real-time evaluation of evidence and identification of strategies for managing virological failure among second-line ART patients in resource-limited settings remain critical challenges, with limited documented information available. This study assessed the magnitude of virological failure among people living with HIV (PLHIV) on second-line ART, aged 18 years and older, enrolled from 2018 to 2022 in Addis Ababa, Ethiopia. It was found that 14.9% (55/369) of the total study subjects had virological failure. Even though the study design was somehow different from others, it was concordance with a

study with an overall prevalence rate of VF of 15.4% in Ethiopia [26] and was done in Pune, India, with a virological failure of 15% (59/400) of the study participants [27].

The finding is also significantly higher than the rates observed in other studies with a VF of 12.2 in Ethiopia [6] which is more concordance and supported by previous studies carried out in a similar sub-Saharan countries, with in Rwanda, virological failure was 12% for 26 months [28], and in Tanzania, a study reported a 12.18% prevalence of virological failure among participants on second-line ART over six months of follow-up conducted by Gunda et al. in 2019 [29]. However, it was found far from the recent WHO-recommended target, which sets a threshold for virological failure (VF) at less than 10% [30]. and also the 95% UNAIDS viral suppression target by 2025 [10].

Moreover, the findings of this study revealed that it is slightly lower than compared of a previous similar study done in Ethiopia with a pooled prevalence of virological failure of 15.95% done by Aytenew [31], and a study conducted in Tanzania reported a virological failure rate of 29.72% over a two-year follow-up period [32].

This was also found slightly inferior to a study done in China, with a cumulative rate of virological failure were 18.45% for the high/low-level viremia group [33], and a cohort study conducted in Malawi showed that 32% of participants experienced virological failure while on second-line ART [34].

The variability difference across all magnitudes of VF in these studies may be attributed to differences in study designs, which might arise from disparities in health system-related gaps, variations in country context and development, urban complexities, population characteristics, methodological approaches, and study settings.

Virological failure in an HIV treatment regimen occurs when individuals living with HIV are unable to suppress viral replication despite starting treatment, or they could rebound after an initial suppression. This failure can lead to increased risks of disease progression, medication toxicity, and drug resistance. It is often associated with risk predictors that contribute to high rates of virological failure.

In this context, several factors were found to be significantly associated with VF, including poor adherence, noncompliance with medical care, lost to follow-up, transfer in from other health facilities for follow-up, having a history of frequent regimen changes, and drug toxicity drawn from the qualitative and quantitative data sets.

Adherence is defined as 'the act of following the provider's instructions regarding the timing, dosage, and frequency of medication intake' [17]. Similarly, a study done by Aytenew et al [31], and it's supported by another similar study conducted by Opoku in Ghana [35], which is concordant with this research finding as clients with poor adherence to ART present a significant risk factor for VF, with an AOR of 6.641 (95% CI: 1.077, 40.95, P-value: 0.041). Additionally, our qualitative findings supported this as well.

Similarly, the current study found that individuals who were transferred from other health facilities for treatment follow-up were 1.97 times more likely to have a high viral load and at a higher risk of virological failure than those who had care at their original enrolled health facility. This could be due to a lack of continuity of follow-up clinical monitoring, unformed patients to clinical provider relationship, communication gaps b/n the referring and referee health facilities. Perceived patients' safety and comfort variability in care quality, and logistic and related administrative factors and which could increase the potential of treatment disruption and worsen the virological failures [36,37].

Thus, transferred-in patients were prone to related challenges that elevate the risk of virological treatment failure. Having standardized referral systems, enhancing digitalizing record-sharing systems, and providing robust individualized adherence support are some of the most essential prioritized activities for improving treatment outcomes.

Another risk factor for non-suppressing VF among ART patients was lost follow-up. Patients who are lost to follow-up (LTFU) from their HIV care while on second-line ART face a significantly higher risk of virological failure, with an adjusted hazard ratio (AHR) of 2.52 (95% CI: 1.35–4.69, p-value = 0.004), compared to those who adhere consistently to their treatment care. Evidence from Ethiopia and Nigeria [38,39] and South Africa further highlights LTFU as a key contributing factor to virological non-suppression [40]. This underscores the importance of maintaining comprehensive follow-up care to minimize virological failure and improve patient outcomes.

This could lead to drug resistance, mostly because of frequent interruption or on and off, interrupted adherence, missed opportunities for timely drug and treatment adjustments, and loss of re-engagement. Hence, to overcome this, a proactive retention strategy, targeted and differential service modalities for the strong re-instatement and engagement in the treatment follow-up are very crucial, as indicated in this research qualitative study participants highlighted as well.

Patients who have experienced regimen change of their second-line ART follow-up are twice as likely (AHR: 2.05; 95% CI: 1.08–3.88) to experience virological treatment failure compared to clients with fewer regimen change experiences. This finding was supported by a similar study conducted in China found that individuals who modified their ART regimen were 1.728 times more likely to exhibit a high viral load than those who adhered to their original treatment regimen, which is supported by our qualitative data as well [33].

This could be psychological fatigue, cross-resistance with residue of drugs in absorption by the client's circulation system, loss of trust in and demoralization of the success of the treatment, which could lead to cumulative drug-related treatment failure vulnerabilities.

Regimen changes during ART follow-up may lead to virological failure, while drug interactions can affect drug levels, efficacy, and toxicity. Addressing adherence, monitoring interactions, and selecting appropriate regimens are key to preventing second-line ART failure. Continuous monitoring and personalized care are essential for better outcomes [41].

Thus, identifying and timely mitigating these contributing risk factors was at most important to improve the PLHIV quality of life.

## Strengths and limitations of the study

The study was conducted in a hospital setting where the routine ART program was active, engaging professionals and capturing patients' real-world experiences. It employed a mixed-methods design, integrating both quantitative and qualitative approaches, to examine the prevalence of VF and its associated risk factors. This methodology addresses pressing public health concerns in Ethiopia and contributes to filling gaps in the existing literature.

Despite these strengths, the study has limitations that warrant careful consideration when interpreting and applying its findings. Due to limited capacity for drug resistance testing, the specific factors of VF could not be fully explored. Additionally, low-level viremia was not measured; viral loads were categorized using a 1,000 copies/mL threshold, primarily due to resource constraints. We recommend that future research address these issues.

## Conclusion

ART failure among HIV-positive adults is a significant public health issue, particularly in resource-limited settings, as it can lead to increased mortality and drug resistance. Virological failure of second-line treatment poses challenges in managing HIV infections and may force patients to seek more expensive alternatives, negatively impacting their health and quality of life.

A study in Addis Ababa, Ethiopia, analyzed second-line virological failure in 369 study participants, revealing that 14.9% developed second-line treatment failure during a total of 20,187 person-months of analysis. This rate of treatment failure is notably higher than the WHO-recommended target of less than 10%. The overall incidence density was 27.2 per person-month (PM), or 33 per 1,000 person-years, highlighting the need for careful monitoring and adjustments to treatment plans.

Key risk factors identified include poor adherence, loss to follow-up, frequent regimen changes, perceived stigma, and drug toxicity.

The research highlights the significance of the Enhanced Adherence Counseling (EAC) program in achieving viral suppression among second-line ART clients. Given the limited treatment options, active patient monitoring and timely treatment adjustments are crucial to improve quality of life and resource utilization. To mitigate the financial and logistical burden caused by stigma-driven long-distance travel by immediately decentralizing HIV care into local, and implementing multi-month dispensing (MMD) for stable patients to minimize travel frequency.

Regular virological monitoring, prompt switches upon identifying treatment failures, and closer follow-up for high-risk patients were some of the identified key recommended points. Additionally, further studies on viremia and genomic-based drug resistance in those experiencing virological failure are strongly encouraged.

## Supporting information

**S1 Fig. Concurrent mixed cohort research design used in this study.**
(DOCX)

**S2 Fig. Infographic depicting the overall summary of defining virological failure in the research process.**
(DOCX)

**S3 Fig. A time range by which the patients developed OI while on second-line ART in Addis Ababa, Ethiopia.**
(DOCX)

## Acknowledgments

We are very grateful and acknowledge the UNISA, Addis Ababa University, Addis Ababa Health Bureau, the Zewditu Memorial Hospital, and Yekatit 12 Hospital Medical College for granting their institutional ethical approval and their support and assistance in accessing diverse resources used in the study.

We would like to express our appreciation and thanks to Mr. Enkusilasie Tegegn for his technical assistance in designing the infographic summarization.

We also acknowledge all the supervisors, data collectors, and study participants.

## Author contributions

**Conceptualization:** Bekelech Bayou Feyissa, Eugene Lee Davids.

**Data curation:** Bekelech Bayou Feyissa, Abay Sisay, Anteneh Yalew, Geoffrey Setswe.

**Formal analysis:** Bekelech Bayou Feyissa, Abay Sisay, Anteneh Yalew, Geoffrey Setswe.

**Funding acquisition:** Bekelech Bayou Feyissa, Abay Sisay.

**Investigation:** Bekelech Bayou Feyissa, Abay Sisay, Eugene Lee Davids, Geoffrey Setswe.

**Methodology:** Bekelech Bayou Feyissa, Abay Sisay, Eugene Lee Davids, Anteneh Yalew, Geoffrey Setswe.

**Project administration:** Bekelech Bayou Feyissa, Abay Sisay, Geoffrey Setswe.

**Resources:** Bekelech Bayou Feyissa, Abay Sisay.

**Software:** Bekelech Bayou Feyissa, Abay Sisay, Eugene Lee Davids, Anteneh Yalew, Geoffrey Setswe.

**Supervision:** Bekelech Bayou Feyissa, Eugene Lee Davids, Anteneh Yalew, Geoffrey Setswe.

**Validation:** Bekelech Bayou Feyissa, Abay Sisay, Anteneh Yalew, Geoffrey Setswe.

**Visualization:** Bekelech Bayou Feyissa, Abay Sisay, Anteneh Yalew, Geoffrey Setswe.

**Writing – original draft:** Bekelech Bayou Feyissa, Abay Sisay, Eugene Lee Davids, Anteneh Yalew, Geoffrey Setswe.

**Writing – review & editing:** Bekelech Bayou Feyissa, Abay Sisay, Eugene Lee Davids, Anteneh Yalew, Geoffrey Setswe.

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
