## [Decision Letter · Decision Letter 0]

30 Sep 2025

Dear Dr. Feyissa,

Thank you for submitting your manuscript to PLOS ONE. After careful consideration, we feel that it has merit but does not fully meet PLOS ONE’s publication criteria as it currently stands. Therefore, we invite you to submit a revised version of the manuscript that addresses the points raised during the review process.

The reviewers appreciated the work and amount of data generated but have made several suggestions for improvement, including optimising the data to show and how they are presented. Kindly consider all the reviwers comments in revising the manuscript.

We look forward to receiving your revised manuscript.

Kind regards,

Chika Kingsley Onwuamah, Ph.D.

Academic Editor

PLOS ONE

Journal Requirements:

4. We note that Figure 2 in your submission contain map/satellite images which may be copyrighted. All PLOS content is published under the Creative Commons Attribution License (CC BY 4.0), which means that the manuscript, images, and Supporting Information files will be freely available online, and any third party is permitted to access, download, copy, distribute, and use these materials in any way, even commercially, with proper attribution. For these reasons, we cannot publish previously copyrighted maps or satellite images created using proprietary data, such as Google software (Google Maps, Street View, and Earth). For more information, see our copyright guidelines: http://journals.plos.org/plosone/s/licenses-and-copyright.

1. You may seek permission from the original copyright holder of Figure 2 to publish the content specifically under the CC BY 4.0 license.

Reviewers' comments:

Reviewer's Responses to Questions

**Comments to the Author**

1. Is the manuscript technically sound, and do the data support the conclusions?

Reviewer #1: Yes

Reviewer #2: Partly

2. Has the statistical analysis been performed appropriately and rigorously?

Reviewer #1: Yes

Reviewer #2: I Don't Know

3. Have the authors made all data underlying the findings in their manuscript fully available?

Reviewer #1: Yes

Reviewer #2: Yes

4. Is the manuscript presented in an intelligible fashion and written in standard English?

Reviewer #1: Yes

Reviewer #2: No

Reviewer #1: I recommend that the manuscript undergo minor revisions before it is ready for publication. The authors should address comments and suggestions below to improve the manuscript.

It should address the below comments:

1. Quantitative results: The abstract reports the prevalence of virological failure as 14.9% and the incidence density as 27.2 per 10,000 person-months. It would be helpful also to report the confidence intervals for these estimates to provide a sense of their precision.

2. Qualitative results: The methods section provides a good overview of the quantitative methodology, but has less detail on the qualitative methodology. The authors should provide more information on the following:

a. Sampling: How were the participants for the KIIs and FGDs selected? What were the specific inclusion and exclusion criteria?

b. Data Collection: How were the KIIs and FGDs conducted? What were the key topics covered in the interview and discussion guides?

c. Data Analysis: How was the qualitative data analyzed? What specific thematic analysis approach was used?

d. Mixed-Methods Integration: The methods section mentions that the quantitative and qualitative data were integrated through triangulation, but it does not provide any detail on how this was done. The authors should describe the specific steps they took to integrate the two datasets.

3. The quantitative results are generally well-presented, but the tables and figures could be improved. For example, Table 2 is very dense and difficult to read. The authors should consider splitting this table.

4. Presentation of Qualitative Results: The qualitative results are presented as a series of quotes, which is effective in illustrating the key themes. However, the authors should provide more of their own analysis and interpretation of the quotes, rather than simply presenting them.

5. Integration of Quantitative and Qualitative Results: The results section presents the quantitative and qualitative results separately. The authors should consider integrating the two datasets more closely, for example, by using quotes to illustrate the quantitative findings.

6. Discussion: The discussion section provides a good overview of the key findings, but it could be strengthened by more fully integrating the quantitative and qualitative findings. The authors should discuss how the qualitative findings help to explain the quantitative results.

7. Limitations: The authors acknowledge some of the limitations of the study, but they could be more comprehensive. For example, they should discuss the potential for selection bias in the qualitative component of the study.

Reviewer #2: Thank you for submitting your interesting work.

Unfortunately, as reviewer, I did not find the manuscript coehsive but instead appeared to be an attempt to include all collected data regardless of relevance to the scientific questions at hand.

This paper needs major revisions with an eye towards the following:

1) Determine the most important scientific question regarding 2nd-line ART and limit your analysis to answering that question even if it means foregoing presentation of data that I am sure you have painstakingly collected

2)When presenting your results avoid the use of subjective statements justifying the presentation of your results. Your results should speak for themselves and if elaboration is needed, reserve for the discussion

3) Your introduction should be limited to the key scientific question with background followed by a brief statement justifying your approach to answering the question

4) Limit your figures to 2-3 to amplify your results and justify your conclusions

Also would like to see you show data on prior regimens at the time of switch. Given time period you are reporting on, a reader would need to know if the prior regimen was NNRTI-based or DTG-based. If NNRTI-based, I am not sure this would be very relevant to the DTG-era but you could try to make a case that it is.

**Do you want your identity to be public for this peer review?** For information about this choice, including consent withdrawal, please see our Privacy Policy

Reviewer #1: No

Reviewer #2: No

---

## [Author Response · Author response to Decision Letter 1]

18 Oct 2025

Intro:

We would like to sincerely thank the editor and the reviewers for the time taken to review our work and provide constructive feedback. Your comments have been instrumental in improving the quality of the manuscript. Below, we detail how we have addressed each point as raised by the editor and reviewers.

Response to Journal Requirements.

Response:

We have noted and ensured our revised manuscript complies with the PLOS ONE style template

2. Please note that funding information should not appear in any section or other areas of your manuscript.

Response:

We have noted and removed funding information from the revised manuscript

Response:

Thank you, revised accordingly.

4. We note that Figure 2 in your submission contain map/satellite images which may be copyrighted. All PLOS content is published under the Creative Commons Attribution License (CC BY 4.0), which means that the manuscript, images, and Supporting Information files will be freely available online, and any third party is permitted to access, download, copy, distribute, and use these materials in any way, even commercially, with proper attribution. For these reasons, we cannot publish previously copyrighted maps or satellite images created using proprietary data, such as Google software (Google Maps, Street View, and Earth). For more information, see our copyright guidelines: http://journals.plos.org/plosone/s/licenses-and-copyright.

1. You may seek permission from the original copyright holder of Figure 2 to publish the content specifically under the CC BY 4.0 license

Response:

Thanks for raising the copyright issue once again regarding Figure 1 (in PONE-D-25-41462R1 version) of the manuscript. Although the figure was not adopted from an online map/ satellite image. Rather, the map was originally constructed using the ArcMap 10.4.1 software from the available shape files of Ethiopia. Just for simplicity and to make a more convenient and comply with the editor’s prior suggestion, and to avoid back and forth, we decided and removed the figure from the manuscript.

Response:

Thank you, and noted.

Response to reviewer comments.

Reviewer #1

1. Quantitative results: The abstract reports the prevalence of virological failure as 14.9% and the incidence density as 27.2 per 10,000 person-months. It would be helpful also to report the confidence intervals for these estimates to provide a sense of their precision.

Response:

We thank the reviewer for this suggestion. We have added the 95%CI and read as “14.9%, (95% CI 11, 19), lines 31 to 32, lines 324 and 340 of the revised clean version of the manuscript.

2. Qualitative results: The methods section provides a good overview of the quantitative methodology, but has less detail on the qualitative methodology. The authors should provide more information on the following:

a. Sampling: How were the participants for the KIIs and FGDs selected? What were the specific inclusion and exclusion criteria?

Response:

We thank the reviewer for this suggestion. “Both KII and FGD participants were selected purposefully. The KII participants were selected based on their status as ART service users, and the FGD participants were selected based on their specific professional roles as HIV service providers in the hospitals and program managers.

Adult PLHIV aged 18 years and older, who were currently taking second-line ART, had a regular follow-up visit to the hospitals during the data collection period, and willingly provided their informed consent for the interview, have been included in KII. Individuals who were on first-line ART, under 18 years old, or who did not volunteer were excluded from the KII”.

Moreover, HIV service providers (including ART clinicians, pharmacists, case managers, data clerks, and adherence supporters at selected hospitals) and HIV program managers at the regional and sub-city health bureau level were included in the FGD. Individuals who did not hold one of these specific roles were excluded from the FGDs. Lines 142 to 153 of the revised version of the manuscript.

b. Data Collection: How were the KIIs and FGDs conducted? What were the key topics covered in the interview and discussion guides?

Response:

To ensure consistency, KII and FGD guiding questions were developed, and all data collectors were oriented on these questions before data collection commenced, ensuring the information was gathered accordingly.

The key topics covered during the interviews and discussions centered on challenges, developing effective and targeted strategies to improve ART adherence and achieve optimal health outcomes for PLHIV taking second-line ART. Lines 213 to 218 of the revised version of the manuscript.

c. Data Analysis: How was the qualitative data analyzed? What specific thematic analysis approach was used?

Response:

The qualitative data analysis was conducted using a thematic analysis approach, by Atlas.ti software, Version 24. The qualitative data from the KIIs and FGDs were imported into Atlas.ti software, where a systematic thematic analysis was conducted by grouping the documents, generating initial codes, refining and grouping the coded segments, and then performing final analysis and visualization. Hence, the themes were generated directly from the data without trying to fit them into pre-existing theory or framework. The specific theme the researcher used was the inductive thematic analysis. Lines 238 to 244 of the revised version of the manuscript.

d. Mixed-Methods Integration: The methods section mentions that the quantitative and qualitative data were integrated through triangulation, but it does not provide any detail on how this was done. The authors should describe the specific steps they took to integrate the two datasets.

Response:

We thank the reviewer for this suggestion. We used the following general steps: Independent analysis, conversion, and comparison, point-by-point triangulation, and integrated interpretation.

The quantitative and qualitative data were analyzed separately using SPSS software version 28 and Atlas ti software version 24, respectively. Key findings from both datasets were converted into a common format. Quantitative results were presented as statistical outcomes, while qualitative themes were converted into descriptive statements. The results were systematically compared on a point-by-point basis to identify areas of convergence, divergence, and expansion. The final discussion and conclusion synthesized these compared findings. Lines 248 to 254. Moreover, it was illustrated in S file 1 (line 116), in the revised version of the manuscript.

3. The quantitative results are generally well-presented, but the tables and figures could be improved. For example, Table 2 is very dense and difficult to read. The authors should consider splitting this table.

Response:

We thank the reviewer for the thoughtful suggestion. We fully appreciate the importance of clarity and readability, especially when presenting dense quantitative data. In this case, however, we intentionally structured the table as a single, continuous unit to preserve the integrity and coherence of the dataset. Splitting it into separate tables may risk fragmenting the narrative and obscuring key relationships across variables. We have carefully reviewed its layout and ensured that it remains accessible while conveying the full message to readers who seek comprehensive insight. With due respect to the reviewer’s suggestion, we believe maintaining it as one unified table could best serve the analytical purpose of the manuscript.

4. Presentation of Qualitative Results: The qualitative results are presented as a series of quotes, which is effective in illustrating the key themes. However, the authors should provide more of their own analysis and interpretation of the quotes, rather than simply presenting them.

Response:

We thank the suggestion. We added some analysis on the selected quotes, as indicated in the revised version of the manuscript.

5. Integration of Quantitative and Qualitative Results: The results section presents the quantitative and qualitative results separately. The authors should consider integrating the two datasets more closely, for example, by using quotes to illustrate the quantitative findings.

Response:

We appreciate the thoughtful suggestion. In our study, we employed both quantitative (survey-based) and qualitative (KII and FGD) methodologies, each designed to address distinct research questions. The survey data specifically explored the following areas:

• Reasons for switching to second-line ART regimens

• opportunistic infections among individuals on second-line ART

• Time to OI in PLHIV receiving second-line ART

• Magnitude and associated risk factors for virological failure (VF) in second-line ART

• Time to treatment failure among patients on second-line ART

To enhance clarity and alignment, we have re-labeled the qualitative: KII and FGD results to reflect the two additional thematic areas under investigation:

• Reasons for virological failure derived from key informant interviews

• Factors influencing virological failure in second-line ART based on focus group discussions

Final interpretation was achieved through a synthesis of both data streams, enabling mutual validation and deeper contextual understanding. For example, the survey revealed a statistically significant association between non-disclosure of HIV status and increased risk of virological failure. This quantitative finding was strongly reinforced by focus group participants, including hospital-level ART providers and program managers, who emphasized that non-disclosure often limits access to social support, thereby undermining medication adherence and negatively affecting treatment outcomes.

6. Discussion: The discussion section provides a good overview of the key findings, but it could be strengthened by more fully integrating the quantitative and qualitative findings. The authors should discuss how the qualitative findings help to explain the quantitative results.

Response:

We appreciate the suggestion, and the recent discussion of results was mainly derived from the triangulation of qualitative and quantitative datasets. This is addressed, as indicated (for sample) in lines 638-641, 647,667, 672-673, 679, and 683-687 in the revised version of the manuscript.

7. Limitations: The authors acknowledge some of the limitations of the study, but they could be more comprehensive. For example, they should discuss the potential for selection bias in the qualitative component of the study.

Response:

Thank you for highlighting this important point. We fully recognize the potential for selection bias in qualitative research and appreciate your suggestion to address it more explicitly. In response, we have made revisions to the Methods section of the revised manuscript to reflect this concern. Specifically, we now clarify that participant selection was guided by a clearly defined inclusion criterion, applied consistently by a trained data collector under close supervision to minimize bias. We have added the following statement to the Methods section (lines 223–225) of the revised manuscript: “To mitigate potential selection bias, participants were recruited based on a pre-tested standardized inclusion criterion by a trained research data collector, with oversight from the study supervisor to ensure procedural fidelity.”

Reviewer #2:

Thank you for submitting your interesting work. Unfortunately, as reviewer, I did not find the manuscript coehsive but instead appeared to be an attempt to include all collected data regardless of relevance to the scientific questions at hand.

This paper needs major revisions with an eye towards the following:

1) Determine the most important scientific question regarding 2nd-line ART and limit your analysis to answering that question even if it means foregoing presentation of data that I am sure you have painstakingly collected

Response:

We thank the reviewer for this insightful and constructive recommendation. We fully acknowledge the importance of focusing the manuscript on a clearly defined and impactful scientific question. Accordingly, we have refined the scope of our analysis to emphasize the magnitude of virological failure and its most significant risk factors among people living with HIV receiving second-line ART in Addis Ababa, Ethiopia. This central question is both timely and critical for informing targeted public health interventions, optimizing resource allocation for second- and third-line ART, and guiding clinical decision-making for high-risk populations.

While we recognize the value of brevity in scientific reporting, we believe that the inclusion of both quantitative and qualitative data analyzed concurrently through a mixed cohort design (lines 109–110) enhances the depth and applicability of our findings. In line with this, our research result seems bulk, but as clarified in the revised manuscript (lines 101–106), this approach allows for a more nuanced understanding of the contextual factors influencing treatment outcomes, which we consider essential for policymakers, program managers, and researchers working to strengthen ART programs.

We have carefully reviewed and streamlined the presentation of results to maintain clarity and relevance, while preserving the integrity of the evidence base. We trust that this focused revision aligns with the reviewer’s guidance and strengthens the manuscript’s scientific contribution.

2)When presenting your results avoid the use of subjective statements justifying the presentation of your results. Your results should speak for themselves and if elaboration is needed, reserve for the discussion

Response:

We thank the suggestion, and to the best of our capacity, we revised the result to be more objective by avoiding the subjective nature of the result findings

3) Your introduction should be limited to the key scientific question with background followed by a brief statement justifying your approach to answering the question

Response:

We thank the suggestion, we refined it accordingly

4) Limit your figures to 2-3 to amplify your results and justify your conclusions

Also would like to see you show data on prior regimens at the time of switch. Given time period you are reporting on, a reader would need to know if the prior regimen was NNRTI-based or DTG-based. If NNRTI-based, I am not sure this would be very relevant to the DTG-era but you could try to make a case that it is

Response:

We sincerely appreciate the reviewer’s thoughtful feedback and constructive suggestions.

We tried to minimize the number of figures to 4 to present our results based on the stated research

---

## [Decision Letter · Decision Letter 1]

2 Nov 2025

Dear Dr. Feyissa,

**The reviewers agree attempts have been made to revise the manuscript. However, they have additional comments and suggestions for improvement. Kindly pay attention to the issues about mixed methods raised by the second reviewer in previous submission. This aspect still need clarification.**

We look forward to receiving your revised manuscript.

Kind regards,

Chika Kingsley Onwuamah, Ph.D.

Academic Editor

PLOS ONE

**Journal Requirements:**

Reviewers' comments:

Reviewer's Responses to Questions

**Comments to the Author**

Reviewer #1: All comments have been addressed

Reviewer #2: (No Response)

2. Is the manuscript technically sound, and do the data support the conclusions?

Reviewer #1: Yes

Reviewer #2: No

3. Has the statistical analysis been performed appropriately and rigorously?

Reviewer #1: Yes

Reviewer #2: I Don't Know

4. Have the authors made all data underlying the findings in their manuscript fully available?

Reviewer #1: Yes

Reviewer #2: Yes

5. Is the manuscript presented in an intelligible fashion and written in standard English?

Reviewer #1: Yes

Reviewer #2: No

**Reviewer #1:**  Minor comments to be addressed:

1. In the 'Strengths and limitations of the study' section, authors should check for duplications. Review lines 734-761 of the clean copy and address redundant text.

2. Check inconsistencies in figure numbering. For example figures 24A and B and figure 35 seem to be out of sequence.

3. While the explanation for not splitting table 2 is accepted, it is still challenging in terms of readability . Improving its layout may be helpful.

4. A final proofread will be useful to check consistency in the terminology. For example 'second line' vs '2nd line'

**Reviewer #2:** (No Response)

**Do you want your identity to be public for this peer review?** For information about this choice, including consent withdrawal, please see our Privacy Policy

Reviewer #1: No

Reviewer #2: No

---

## [Author Response · Author response to Decision Letter 2]

9 Nov 2025

9 November 2025

The Academic Editor, PLOS ONE

SUBMISSION OF Re-REVISED MANUSCRIPT: PONE-D-25-41462R1

Attached is the re-revised version of our manuscript entitled “Virological failure and risk factors among people living with HIV taking second-line ART in Addis Ababa, Ethiopia”. We thank you once again for giving us improve the manuscript. All comments and suggestions by the editor and reviewers have been addressed as outlined within this document in a point-by-point fashion, and all changes to the manuscript are indicated with “track changes”.

Please don't hesitate to contact us if you have any questions. All correspondence regarding the manuscript should be addressed to Bekelech Bayou, at bekelechbayou@gmail.com.

Sincerely,

Bekelech B. Feyissa

Addis Ababa,

Ethiopia

Intro:

We would like to sincerely thank once again the editor and the reviewers for the time taken to re-review our work and provide constructive feedback. Your comments have been instrumental in improving the quality of the manuscript. Below, we detail how we have addressed each point as raised by the editor and reviewers.

Editor’s suggestion

• The reviewers agree attempts have been made to revise the manuscript. However, they have additional comments and suggestions for improvement. Kindly pay attention to the issues about mixed methods raised by the second reviewer in previous submission. This aspect still need clarification.

Author Response:

Thank you very much for sharing the issues and the positive suggestions. We appreciate the reviewers’ continued engagement and the editor’s guidance. To clarify our mixed methods approach, we conducted independent quantitative and qualitative analyses using SPSS v28 and Atlas.ti v24, respectively. Key findings from quantitative and qualitative were themed point-by-point to identify convergence, divergence, and expansion, culminating in an integrated interpretation (lines 245–256; S file 1).

The revised manuscript emphasizes how this concurrent mixed cohort design enhances contextual understanding of research findings. We streamlined the presentation for clarity while preserving analytical depth.

Quantitative data addressed treatment switching, opportunistic infections, and virological failure. Qualitative data (KII and FGD) were re-labeled to reflect thematic alignment, focusing on reasons and factors influencing virological failure (lines 383 and 498). For instance, the statistical link between HIV non-disclosure and virological failure was validated by qualitative insights from ART providers and program managers, highlighting its impact on adherence and social support.

Moreover, the recent discussion of results was mainly derived from the triangulation of qualitative and quantitative datasets. This is addressed, as indicated (for sample) in lines 640, 646,667-672, 673-678, and 682-686 in the revised version of the manuscript.

We trust these revisions address the concerns raised and strengthen the manuscript’s methodological clarity and scientific contribution

Reviewers' comments:

Reviewer #1: Minor comments to be addressed:

1. In the 'Strengths and limitations of the study' section, authors should check for duplications. Review lines 734-761 of the clean copy and address redundant text.

Author Response:

Thank you very much for raising the points. We thought the reviewer used the tracked change version of the manuscript (Lines 734-761). However, it lies b/n lines 691-703 in the Clean Version of the previously submitted version.

We conducted a thorough review, avoiding redundant content and making necessary revisions as indicated in lines 689- 700 of the recent revised version of the manuscript.

2. Check inconsistencies in figure numbering. For example figures 24A and B and figure 35 seem to be out of sequence.

Response:

Thank you for highlighting this. The apparent discrepancy may have arisen from referencing the tracked-changes version of the manuscript. For your reference, what was previously labeled as Figure 4A was revised to Figure 2A, and Figure 5 was renumbered as Figure 3 in the previously submitted version. These adjustments were accurately reflected in the clean version of the manuscript, which was submitted to clearly present all revisions made in response to reviewer and editor feedback.

Nonetheless, we have carefully reviewed the figure numbering again and confirm that all figures are now correctly labeled and presented in the appropriate sequence.

3. While the explanation for not splitting table 2 is accepted, it is still challenging in terms of readability . Improving its layout may be helpful.

Response:

Thank you for understanding our concern and allowing us to maintain it as it is, and we did an improvement on the layout of the table as much as possible to increase more e readability.

4. A final proofread will be useful to check consistency in the terminology. For example 'second line' vs '2nd line'

Authors Response:

Thank you for Thank you for the helpful suggestion. The senior authors have conducted a final round of proofreading to ensure grammatical accuracy and consistency in terminology throughout the manuscript. As part of this process, instances of “2nd line” were revised to “second-line” to maintain uniformity in expression.

Reviewer #2: (No Response)

Authors Response:

No specific comments

…………………………………………,,,,,,,,,,……………………………………….

---

## [Decision Letter · Decision Letter 2]

19 Nov 2025

Virological failure and risk factors among people living with HIV taking second-line ART in Addis Ababa, Ethiopia

PONE-D-25-41462R2

Dear Dr. Feyissa,

We’re pleased to inform you that your manuscript has been judged scientifically suitable for publication and will be formally accepted for publication once it meets all outstanding technical requirements.

Kind regards,

Chika Kingsley Onwuamah, Ph.D.

Academic Editor

PLOS ONE

Additional Editor Comments (optional):

Reviewers' comments:

Reviewer's Responses to Questions

**Comments to the Author**

Reviewer #1: All comments have been addressed

2. Is the manuscript technically sound, and do the data support the conclusions?

Reviewer #1: Yes

3. Has the statistical analysis been performed appropriately and rigorously?

Reviewer #1: Yes

4. Have the authors made all data underlying the findings in their manuscript fully available?

Reviewer #1: Yes

5. Is the manuscript presented in an intelligible fashion and written in standard English?

Reviewer #1: Yes

Reviewer #1: I have further reviewed the manuscript and find that the comments from the previous reviews have been addressed.

**Do you want your identity to be public for this peer review?** For information about this choice, including consent withdrawal, please see our Privacy Policy

Reviewer #1: No

---

## [Editor Report · Acceptance letter]

PONE-D-25-41462R2

PLOS One

Dear Dr. Feyissa,

I'm pleased to inform you that your manuscript has been deemed suitable for publication in PLOS One. Congratulations! Your manuscript is now being handed over to our production team.

Kind regards,

on behalf of

Dr. Chika Kingsley Onwuamah

Academic Editor

PLOS One